# Elevation-Dependent Changes to Plant Phenology in Canada's Arctic Detected Using Long-Term Satellite Observations

**Wenjun Chen** [1,*], **Lori White** [2], **Sylvain G. Leblanc** [1], **Rasim Latifovic** [1] and **Ian Olthof** [1]

1    Canada Centre for Remote Sensing, Natural Resources Canada, 560 Rochester St.,
     Ottawa, ON K1A 0E4, Canada; sylvain.leblanc@canada.ca (S.G.L.); rasim.latifovic@canada.ca (R.L.);
     ian.olthof@canada.ca (I.O.)
2    Wildlife Landscape Science Directorate, Environment and Climate Change Canada, 1125 Colonel by Dr.,
     Ottawa, ON K1A 0H3, Canada; lori.white2@canada.ca
*    Correspondence: wenjun.chen@canada.ca; Tel.: +1-(613)-759-7895

**Abstract:** Arctic temperatures have increased at almost twice the global average rate since the industrial revolution. Some studies also reported a further amplified rate of climate warming at high elevations; namely, the elevation dependency of climate change. This elevation-dependent climate change could have important implications for the fate of glaciers and ecosystems at high elevations under climate change. However, the lack of long-term climate data at high elevations, especially in the Arctic, has hindered the investigation of this question. Because of the linkage between climate warming and plant phenology changes and remote sensing's ability to detect the latter, remote sensing provides an alternative way for investigating the elevation dependency of climate change over Arctic mountains. This study investigated the elevation-dependent changes to plant phenology using AVHRR (Advanced Very High Resolution Radiometer) time series from 1985 to 2013 over five study areas in Canada's Arctic. We found that the start of the growing season (SOS) became earlier faster with an increasing elevation over mountainous study areas (i.e., Sirmilik, the Torngat Mountains, and Ivvavik National Parks). Similarly, the changes rates in the end of growing season (EOS) and the growing season length (GSL) were also higher at high elevations. One exception was SOS in the Ivvavik National Park: "no warming trend" with the May-June temperature at a nearby climate station decreased slightly during 1985–2013, and so no elevation-dependent amplification.

**Keywords:** elevation dependency; plant phenology; growing season; remote sensing; Arctic mountains

## 1. Introduction

Since the industrial revolution, air temperatures in the Arctic have increased at almost twice the global average rate [1,2]. On top of the warming trend, some studies showed the dependence of surface warming on elevation, with greater warming rates at higher altitudes, namely, the elevation dependency of climate change [3–6]. This elevation-dependent climate change could have important implications for the fate of glaciers and ecosystems at high elevations under climate change. Many rivers in the world are fed by the melting snow and ice of glaciers. The freshwater from glaciers also forms some rare "polar oases" in the high Arctic, such as the wetlands of Bylot Island, Nunavut, Canada. These wetlands have exceptional productivity for an Arctic ecosystem and attract and sustain a wide variety of breeding migratory bird species, including herbivores such as the cackling goose, the rock ptarmigan, and the greater snow goose (http://www.cen.ulaval.ca/bylot/en/bylotstudysite.php/, accessed on 20 August 2021). If the elevation dependency of climate change holds, these glaciers can melt and disappear under a changing climate at an even higher rate (https://nsidc.org/cryosphere/glaciers/questions/climate.html/, accessed on 20 August 2021, [7]). In turn, the melting of glaciers can feedback positively to climate warming due to the reduction of albedo [7], and dramatically changes the climate system

because of the ocean current alternations such as the collapse of the Atlantic Meridional Overturning Circulation [8].

Two types of climatic elevation dependency have been reported (e.g., [3,4,9–13]). Type 1 is the elevation dependency of a climate variable, referring to the relationship between the mean value of a climate variable and elevation. The other type is the elevation dependency of a climate change signal, referring to the relationship between the long-term rate of change of a climate variable and elevation. Examples of type 1 climatic elevation dependency include decreased surface temperature with elevation [1] or mean snow depth with elevation towards middle slope followed by a decrease at the highest elevations [2]. Due to the increasing importance of climate change, many recent studies focused on type 2 climatic elevation dependency, aiming to answer the question: Did elevation amplify a climate change signal? So far, the conclusions on the elevation dependency of temperature change drawn from climate records have been less consistent. For example, supporting evidence was observed over the Swiss Alps [3] and the Tibetan plateau [4–6]. However, results from the tropical Andes [14] and the North American Rocky Mountains [12,15,16] include supportive and contrary evidence. Similarly, a regional climate model showed a substantial elevation dependency of the simulated temperature change signal over the Alpine region [11,13]. However, global analyses of temperature trends for high elevation regions indicated that the relationship between the magnitude of temperature trends and elevation was increasingly supported by observational evidence but not always significant [17–19].

There are many possible causes for the inconsistency on the elevation dependency of a climate change signal [12], such as the lack of sufficient observation sites where long-term climate records are available for a given mountainous study area. Over the mountainous regions, especially in the Arctic, the density of climate stations is usually very low. Researchers pooled long-term temperature trends from different climatic zones regionally or globally in order to have a large enough sample size so that statistically meaningful analysis between elevation and temperature trends can be conducted. Among these different climate zones, the rates of climate change might differ significantly even without the influences of elevation. Consequently, the effect of elevation could easily be masked by the effect of variation in temperature change rates over different climate zones. Plant phenology is strongly controlled by climate and has consequently become one of the most reliable bioindicators of ongoing climate change [20,21]. For example, plants in the Arctic leafed out consistently within 1–6 d after snow-free [21]. Remote sensing technology has been used widely for detecting long-term changes in plant phenology over the Arctic landmass, such as the start of the growing season (SOS), the end of the growing season (EOS), and the growing season length (GSL). For example, remote sensing data reveal widespread lengthening of the growing season and increased gross primary productivity, also called "greening", associated with warmer air temperatures in the high latitudes during the 1980s and 1990s [22]. Using MODIS (Moderate Resolution Imaging Spectroradiometer) satellite data, Karlsen et al. [23] investigated the spatial and temporal variability in the onset of the growing season on Svalbard, Arctic Norway. Over the Canadian Arctic, Chen et al. [24] developed a biophysically based and objective satellite seasonality observation method (BLOSSOM) for applications over the Arctic. Using this method and the long-term AVHRR (Advanced Very High-Resolution Radiometer) data, Chen et al. [25] found a decoupling between plant productivity and growing season length under a warming climate in Canada's Arctic. They also found that the changes in plant phenology caused by climate change can adversely impact caribou phenology, such as the peak calving date [26].

Because of the linkage between climate warming and plant phenology changes and remote sensing's ability to detect the latter, remote sensing provides an alternative way for investigating the elevation dependency of climate change over Arctic mountains. In addition, remote sensing data have an excellent spatial coverage compared to climate stations which are usually sparse in mountainous areas, especially in the Arctic. Therefore, this study aims to investigate if there are elevation-dependent changes to plant phenology

over mountains in Canada's Arctic, using long-term AVHRR time series from 1985 to 2013. Finally, we will investigate the effect of data pooling on the elevation dependency of a climate change signal, using the consistently processed remote sensing data over different study areas located across Arctic Canada.

## 2. Materials and Methods

### 2.1. Study Areas

The study areas were composed of three mountainous and two relatively flat regions in Canadian Arctic (Figure 1). The three mountainous study areas include the Ivvavik National Park (NP) in the northwest corner of Yukon, the Sirmilik NP in northern Baffin Island and Bylot Island, Nunavut, and the Torngat Mountains NP in Northern Labrador. The British Mountains dominate Ivvavik NP, accounting for about two-thirds of the park's area. The highest elevation in the British Mountains is 1760 m (Table 1). Sirmilik NP has three separate areas: the mountain and upland surrounding Oliver Sound, the rugged plateau of eastern Borden Peninsula, and the mountains and lowlands of Bylot Island. The highest peak is located at the Byam Martin Mountains in Sirmilik NP, with an elevation of 1944 m. As its name implies, Torngat Mountains NP is dominated by the Torngat Mountains, with its highest peak at 1652 m. The two relatively flat study areas are the Wapusk NP in northeast Manitoba and the Bathurst caribou range in Northwest Territories and Nunavut. Within the two relatively flat study areas, elevation varies less than 100 m (Table 1).

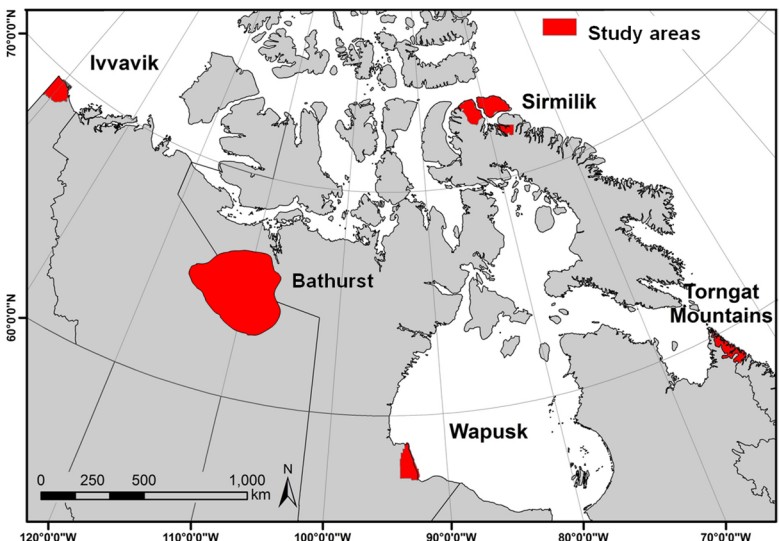

**Figure 1.** The distribution of five study areas in arctic Canada: three mountainous national parks (i.e., Ivvavik, Sirmilik, and the Torngat Mountains) and two relatively flat areas (i.e., Wapusk National Park, and Bathurst caribou summer range and calving ground).

**Table 1.** List of study areas, their attributes, and the closest climate stations that have good data records.

| Study Area | Area (km$^2$) | Latitude/Longitude at the Venter | Ecozone | Elevation Range (m) | Closest Climate Station |
|---|---|---|---|---|---|
| Ivvavik | 9750 | 69°36′00″ N 140°10′00″ W | Taiga cordillera southern Arctic and | 0–1760 | Inuvik |
| Sirmilik | 22,200 | 72°50′4″ N 80°34′55″ W | Arctic cordillera and northern Arctic | 0–1944 | Pond Inlet |
| Torngat | 9700 | 59°22′12″ N 63°38′48″ W | Arctic cordillera | 0–1652 | Nain |
| Wapusk | 11,475 | 57°51′11″ N 93°22′35″ W | Hudson plains | 0–86 | Churchill |
| Bathurst | 112,000 | 65°8′5″ N 111°7′ W | Southern Arctic | 365–465 | Lupin |

These study areas cover five terrestrial ecozones in Canada (Table 1, [27]). They were selected to represent a wide range of northern ecological and elevation conditions. The three mountainous study areas (Ivvavik, Sirmilik, and the Torngat Mountains NPs) are mainly located in the Taiga Cordillera and the Arctic Cordillera. The climate in the Taiga Cordillera Ecozone is extremely cold, humid, with long dark winters and short cool summers. Precipitation averages 250 to 300 mm a year across much of the ecozone. Extensive areas of the alpine tundra occur on the upland plateaus and highest mountain slopes. Scattered Alpine fir trees and a dense understory of willow and shrub birch dominate further downslope. White and black spruce replace firs in the lower parts of this zone. Spruce-lichen woodlands and flat benches of Lodgepole pine dominate below the subalpine. In the lowland, dense spruce-feathermoss forests, riverside communities, and wetlands are also common. In the Arctic Cordillera, the climate is harsh, with long, extremely cold winters and short, cool summers, with only July and August mean daily temperatures above the freezing point. Ice and bald rock dominate 75% of the Arctic Cordillera. Soils are virtually non-existent over much of the area due to ice cover and the slow rate of soil formation. Despite the generally severe conditions, several hardy plant species (e.g., mountain Avens, crustose lichens, cottongrass, Arctic willow, and Arctic white heather) flourish where moisture, heat, and nutrients create favorable microhabitats.

A portion of the Ivvavik NP study area is in the southern Arctic ecozone, whereas some Sirmilik NP is in the northern Arctic ecozone. In addition, the Bathurst caribou range is also located in the southern Arctic ecozone. In the Southern Arctic Ecozone, summer is about four months, whereas winters are long and extremely cold. Total annual precipitation is usually <250 mm in the west and rarely >500 mm in the east. This ecozone is bounded to the south by the treeline. Within the zone, small, scattered clumps of stunted spruce trees grow on warmer, sheltered sites. Low shrubs such as willow, shrub birch, and Labrador tea are well adapted to these conditions. However, on the most exposed sites, low shrubs give way to mats of lichens, mosses, and ground-hugging shrubs such as mountain cranberry and least willow. Summers of the Northern Arctic are short and cold, with mean daily temperatures >0 °C only in July and August. Daily winter temperatures average <−30 °C in the coldest area of this ecozone. Annual precipitation is <250 mm except in southeast Baffin and Labrador, where it can exceed 500 mm. Although much of this region is virtually devoid of plants, relatively lush "oases" are found scattered across the landscape. These oases are confined mainly to coastal lowlands, sheltered valleys, and moist, nutrient-rich corridors along streams and rivers. They often support thick hummocky carpets of sedges, mosses, and lichens and are vital to many wildlife species.

The other flat study area is the Wapusk NP, located in the Hudson plains ecozone, with the average daily temperature ranging from 12 °C to 16 °C in July and −25 °C to −23 °C in January. Average annual precipitation ranges from 500 to 700 mm. Tussocks of sedge, cottongrass, and sphagnum moss dominate the wet areas in the Hudson Plain. Dwarf birch and willow shrubs are also common. On drier sites, shrubby and the low-lying Lapland rosebay, crowberry, blueberry, and cloudberry take hold. South of the tundra is a transition zone known as the taiga. In the lowlands, open stands of White Spruce dominate drier areas, while low stands of willow, black spruce, and tamarack are common on wetter and more exposed sites.

### 2.2. Data Sources

We used 1-km resolution red and near-infrared surface reflectance ($\rho_r$ and $\rho_{nir}$, respectively) and cloud probability from the 10-day AVHRR composites from 1985 to 2013 for quantifying SOS, EOS, and GSL for each land class within a study area. The Canada Centre for Remote Sensing (CCRS) re-processed the AVHRR data sets, including geo-referencing and Bidirectional Reflectance Distribution Function (BRDF) normalization [28], atmospheric correction [29], cloud indexing [30], and inter-sensor normalization [31]. We used the actual acquisition dates within the AVHRR 10-day composites for more accurately estimating SOS and EOS. CCRS paused the re-processing of AVHRR data in 2014 and yet

to resume. Consequently, we could not quantify these growing season variables using AVHRR data since 2014.

Field data of leaf biomass, aboveground biomass, and percentage cover of each vascular plant species were collected at 27 sites in Wapusk National Park during the summer of 2006, at 11 sites in the Ivvavik National Park in 2008, at 16 sites in Torngat Mountains National Park in 2008, at 11 sites in Sirmilik National Park in 2010, and at 34 sites in the Bathurst caribou habitat during 2005, 2013 and 2014. Each site was selected to be relatively homogenous and of a minimum size of 90 m × 90 m [32]. At each site, five to twenty 1-m × 1-m plots were sampled. At each plot, percentage covers of vascular plant species were visually estimated in the field and corrected using digital photos later [33]. All plants were then harvested, identified to species, sorted into dead and live, leaves and stems, and weighted in the field. A sample of these leaves and stems was also taken to the laboratory, oven-dried, and weighed to obtain the oven-dry leaf biomass. The values of leaf biomass and percentage cover at each site were calculated as the average of all plots at the site, and sampling errors were calculated as the standard deviation divided by the square root of sample size.

Other inputs used in the study include land cover or ecotype maps, DEM (Digital Elevation Model) GIS layers, and climate data over the five study areas. The northern land cover map classification using Landsat images by Olthof et al. [34] was used for the Sirmilik NP and Bathurst Caribou habitat. For the Torngat Mountains NP, the northern land cover map missed a portion of this park, so we used an alternative Landsat-derived FGDC land cover [35]. Since wetland classes dominate the Wapusk NP, we developed a wetland map using Li and Chen's method [36]. A more detailed ecotype map was available for the Ivvavik NP [37] and therefore was used in this study. The DEM tiles at 1:50,000 can be downloaded from Geobase (www.geobase.ca/, accessed on 21 September 2020). The climate data (temperature, snow on the ground) are available from Canadian Daily Climate Data (CDCD) (www.climate.weatheroffice.gc.ca/, accessed on 21 September 2020).

### 2.3. Methods

We use a 5-steps approach to derive plant phenology from field measurements and remote sensing data and investigate the elevation dependency.

#### 2.3.1. Determining the Dominant Land Cover Class at 1-km Resolution from Landsat-Based Maps

We aggregated the 30-m Landsat-based land cover map into 1-km resolution so that the results derived from 1-km AVHRR data could be evaluated based on land cover classes. For different AVHRR pixels in a given study area, the purity level—namely, the percentage of the dominant 30-m class pixels in a 1-km pixel—can be different. As a compromise between the purity and the desire to include as many land pixels in a land cover class as possible, we selected >50% purity as inclusion criteria. For example, if the Landsat-based land cover class A, B, C, D, and E composed of 50, 20, 10, 5, and 15% areas, respectively, within a 1-km resolution AVHRR pixel in a given study area, we assigned the pixel as land cover class A. Conversely, if all Landsat-derived classes within a 1-km AVHRR pixel were <50%, we would not include this "unpure" pixel in the analyses.

#### 2.3.2. Constructing Seasonal Profile of Vegetation Index for a Land Cover Class in a Study Area

Despite these extensive pre-processing efforts outlined in the data source section, the 10-d AVHRR composite data can still be very noisy due to residual cloud contaminations and aerosol variations [38]. For example, there could be up to 40% standard error in a vegetation index (e.g., the simple ratio vegetation index (SRVI = reflectance of the near-infrared band/reflectance of the red band) even for a clear-sky pixel due to the effect of aerosol variations. Spatial averaging can effectively reduce this random error caused by aerosol variations by a factor of the square root of the number of pixels averaged [38]. Therefore, a mean vegetation index for a land class in a given study area, usually composed

of many pixels, would be much more accurate than a single pixel. We used SRVI in this study instead of NDVI (Normalized Difference Vegetation Index) because SRVI usually has a linear relationship with leaf biomass over Arctic land ecosystems, whereas NDVI can become nonlinear with leaf biomass and saturated when leaf biomass exceeds about $50 \text{ g m}^{-2}$ [39].

Additionally, a substantial number of pixels for a land cover within a given 10-d composite period could still be contaminated by residual clouds, despite selecting the least cloud contaminated day within the 10-d composite period [30]. Chen et al. [40] developed a method that can effectively reduce these residual cloud contaminations for objectively constructing a seasonal profile of SRVI. In this study, we followed the same procedure, briefly described as follows. AVHRR pixels of a land cover class during a 10-d composite period were divided into four categories: clear sky, lightly cloud contaminated, moderately cloud contaminated, and heavily cloud contaminated. Data pairs of mean SRVI of clear-sky pixels against that for lightly (or medium, or heavily) cloud contaminated pixels were selected for a land cover class from all 10-d composite periods from 1985 to 2013. The relationships between mean SRVI of clear-sky pixels and that of lightly (or medium, or heavily) cloud contaminated pixels for the class were then applied to all cloud contaminated pixels to obtain the estimated clear-sky SRVI. Finally, the seasonal profile of mean SRVI of the land cover class was constructed by averaging over all "pure" pixels in the class with SRVI for cloud-contaminated pixels replaced by the estimated value of clear-sky SRVI.

### 2.3.3. Estimating Seasonal Variations in Leaf Biomass for a Land Cover Class

Leaf biomass for each class at any given time of year can be calculated using the seasonal profile of SRVI once the profile was properly calibrated with field leaf biomass measurements. Because it was difficult to find sites several km across, field measurement sites of leaf biomass were chosen to be at least 90 m by 90 m, in contrast to the 1-km AVHRR spatial resolution. To bridge the scale difference between leaf biomass measurement sites and AVHRR pixels, we used the 30-m resolution Landsat images to scale the field measurements to 1-km AVHRR pixels. We developed a Landsat mosaic for each study area by first selecting a cloud-free middle-summer reference scene and then adding other scenes to the reference using overlapping areas' correlations. There are usually overlapped areas between two adjacent Landsat Scenes. A correlation between pixels within the overlapped area could then be developed and corrected to the value of other scenes to that of the reference scene.

In addition to the spatial scale difference, there could also be a temporal mismatch, in which the date of Landsat SRVI was different from the date of field measurements. To minimize the effect of temporal mismatch, we used daily MODIS data on both dates to correct the Landsat-derived vegetation index values to the middle date of field measurements. MODIS acquires data daily approximately 15 min after Landsat data in the same polar orbit. Linear regressions for red and near-infrared reflectances between Landsat and MODIS were used for the correction of Landsat-derived SRVI from the image acquisition date to the field measurement date.

The corrected Landsat SRVI values were then correlated with field leaf biomass measurement to obtain their relationship. Applying the relationship between Landsat SRVI and leaf biomass to the Landsat mosaic, we produced a baseline map of leaf biomass for each study area. From the baseline map of leaf biomass, we calculated the value of leaf biomass from the date of image acquisition t 1-km spatial resolution by aggregating 30-m pixels into a 1-km AVHRR-resolution pixel. For developing the relationship between AVHRR SRVI and leaf biomass, we selected clear-sky AVHRR pixels at the date of the Landsat mosaic. We then calculated seasonal variations in leaf biomass of each land cover class over each study area by applying the relationship to these SRVI seasonal profiles. The peak leaf biomass was quantified as the highest leaf biomass among all 10-day periods in a growing season.

### 2.3.4. Detecting SOS, EOS, and GSL for a Land Cover Class in a Study Area

The dates of SOS and EOS were estimated using the AVHRR SRVI profile and a threshold defined by the zero deciduous leaf biomass [24]. The exact date of the AVHRR SRVI profile was computed as an average of exact date within the 10-day composite of each pixel used for the profile construction. As such, the threshold was composed of two components: effect of evergreen shrubs that can be quantified by their percentage covers and AVHRR-equivalent SRVI values; and that of all other land cover elements (e.g., bare soil, rock, deciduous plants without green leaves, shadow) that can be calculated by inverting the relationships between leaf biomass and AVHRR SRVI developed using field measurements of leaf biomass [24]. The SOS was determined as the day of the year (DOY) on which the class mean of AVHRR SRVI first became larger than the threshold in the spring, as exemplified in Figure 2.

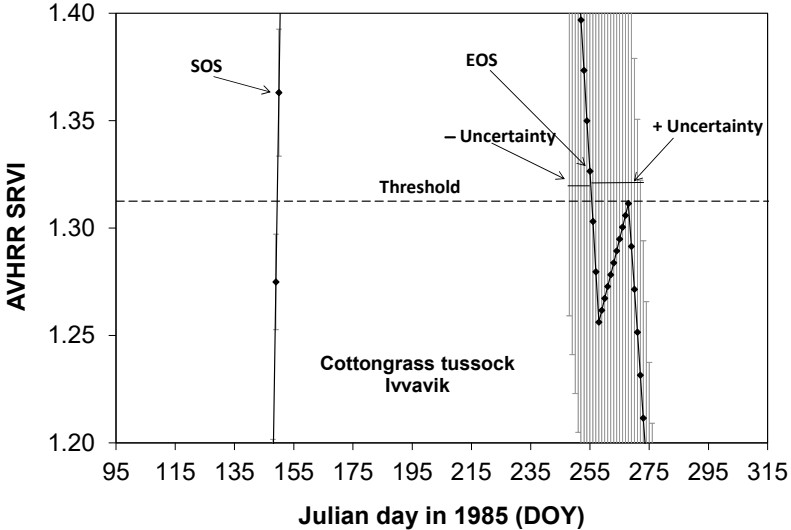

**Figure 2.** A methodological diagram showing the determination of SOS, EOS, and GSL and their uncertainties for the cotton-grass tussock class in the Ivvavik National Park.

Similarly, the EOS was determined as the DOY on which the class mean of AVHRR SRVI last became less than the threshold in the fall. Given that in the Arctic snow could sometimes revisit a vegetated area around SOS or EOS, i.e., there could be multiple crossings of AVHRR SRVI through the threshold. Therefore, only the first crossing with 90% confidence in the spring would be determined as SOS, in which all other days before this date had no value of (the class mean of AVHRR SRVI—its one standard estimation error) was larger than the threshold. Similarly, only the last crossing with 90% confidence in the fall would be determined as EOS.

### 2.3.5. Detecting SOS, EOS, and GSL for a Land Cover Class in a Study Area

The mean elevation of each land cover class was calculated from the DEM data by averaging over all pixels within the class in a given study area. Linear regression was then conducted to investigate the relationships between the trends of these plant phenology dates and other environmental variables (e.g., the mean elevation of the class, the trend in peak leaf biomass).

## 3. Results

### 3.1. Long Terms Trends in SOS, EOS, and GSL during 1985–2013 over the Five Study Areas

Significant GSL trends were found for most tundra classes in the Ivvavik National Park during 1985–2013, as exemplified in Figure 3. For the cotton-grass tussock class, the GSL trend during 1985–2013 was 0.63 d y$^{-1}$, significant at the 90% confidence level (Table 2). In other words, the GSL of the cotton-grass tussock class increased 18.4 days

from 1985 to 2013. Most of the GSL increases of the cotton-grass tussock class were due to the delay in EOS by 14.5 days and significant at the 95% confidence level. On the contrary, the advance in SOS only contributed 3.9 days and was statistically not significant.

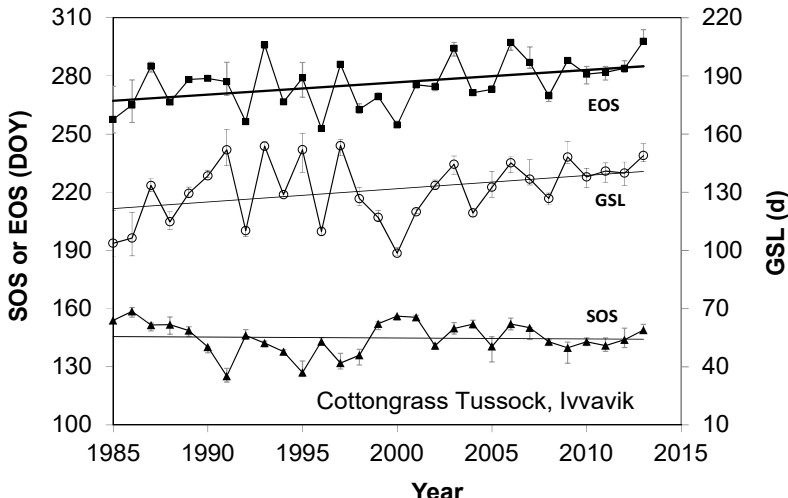

**Figure 3.** Example inter-annual variations and trends in SOS, EOS, and GSL during 1985–2013 for the cotton-grass tussock class in the Ivvavik National Park. Error bars show one standard estimation errors in SOS, EOS, or GSL. Statistics for these trends are listed in Table 2.

**Table 2.** Tundra or wetland class in the five study areas: Ivvavik (denoted with I), the Torgnat Mountains (T), Sirmilik (S), and Wupask (W) National Parks, and the Bathurst (B) caribou range. Also included are long-term trends (i.e., slopes) of SOS, EOS, and GSL during 1985–2013 as well as mean elevation of each class. The superscripts of *, **, and *** stand for significant at the 90%, 95%, and 99% confidence level, respectively.

| Label | Land Cover Class Name | Mean Elevation (M) | Trends (d y$^{-1}$) | | |
|---|---|---|---|---|---|
| | | | SOS | EOS | GSL |
| I1 | Alpine slope | 764 | −0.24 | 0.64 *** | 0.88 ** |
| I2 | Willow-horsetail wet slope | 599 | −0.23 | 0.62 ** | 0.85 ** |
| I3 | Rock lichen | 912 | −0.10 | 0.70 *** | 0.81 * |
| I4 | Willow-birch moist slope | 722 | −0.32 | 0.49 ** | 0.81 ** |
| I10 | Willow floodplain | 215 | 0.01 | 0.57 ** | 0.56 * |
| I18 | Cotton-grass tussock | 241 | −0.13 | 0.51 ** | 0.63 * |
| I20 | Willow-sedge pediment drainage channel | 56 | −0.16 | 0.52 * | 0.66 ** |
| I22 | Sand/silt | 426 | −0.14 | 0.64 ** | 0.79 * |
| I23 | Hedysar-avens inactive alluvial Terrace | 26 | −0.12 | 0.38 | 0.53 |
| I25 | Willow-coltsfoot drainage channel | 183 | −0.24 | 0.45 | 0.65 * |
| I26 | Sedge tussock | 192 | −0.05 | 0.54 ** | 0.59 * |
| S1 | Tussock graminoid tundra | 249 | −0.22 | 0.28 | 0.50 |
| S2 | Wet sedge | 282 | −0.26 | 0.37 | 0.63 * |
| S3 | Moist-dry graminoids/dwarf shrub | 391 | −0.30 | 0.41 * | 0.71 * |
| S7 | Prostrate dwarf shrub | 208 | −0.37 ** | 0.36 | 0.73 ** |
| S8 | Sparsely vegetated bedrock | 379 | −0.37 | 0.57 | 0.94 * |
| S9 | Sparsely vegetated till-colluvium | 474 | −0.45 * | 0.56 * | 1.01 ** |
| S10 | Bare soil/cryptogam crust-frost boils | 398 | −0.43 * | 0.46 ** | 0.89 ** |
| S12 | Barren | 611 | −0.54 * | 0.53 | 1.08 |
| T16 | Deciduous shrub (>75% cover) | 69 | −0.32 | 0.92 *** | 1.27 *** |
| T23 | Herb-shrub | 292 | −0.30 | 1.08 *** | 1.34 *** |
| T24 | Shrub-herb-lichen-bare | 23 | −0.19 | 0.66 ** | 0.84 ** |

**Table 2.** *Cont.*

| Label | Land Cover Class Name | Mean Elevation (M) | Trends (d y⁻¹) | | |
|---|---|---|---|---|---|
| | | | SOS | EOS | GSL |
| T26 | Lichen-shrubs-herb, bare soil, rock outcrop | 267 | −0.45 | 0.90 *** | 1.33 *** |
| T28 | Low veg cover (bare soil, rock outcrop) | 629 | −0.50 | 1.05 ** | 1.57 ** |
| T35 | Lichen barren | 331 | −0.44 | 0.76 *** | 1.20 *** |
| T36 | Lichen-shrub-herb-bare | 325 | −0.37 | 0.80 *** | 1.16 ** |
| T38 | Rock outcrop low vegetation cover | 684 | −0.39 | 1.01 ** | 1.46 * |
| W3 | Dryas heath upland | 3 | −0.24 | 1.02 *** | 1.25 *** |
| W6 | Lichen peat plateau bog | 38 | −0.22 | 0.89 *** | 1.11 *** |
| W9 | Lichen melt pond bog | 48 | −0.34 | 0.96 *** | 1.30 *** |
| W10 | Sedge fen | 5 | −0.34 | 0.97 *** | 1.31 *** |
| W11 | Shrub fen | 13 | −0.44 | 0.91 *** | 1.35 *** |
| W12 | Shrub sedge fen | 9 | −0.22 | 0.94 *** | 1.16 ** |
| B16 | Shrub moist | 365 | −0.33 ** | 0.48 ** | 0.81 *** |
| B17 | Shrub mesic | 370 | −0.34 * | 0.50 ** | 0.84 *** |
| B23 | Herb | 400 | −0.24 | 0.52 *** | 0.76 *** |
| B26 | Lichen-shrubs-herb, bare soil, rock outcrop | 370 | −0.28 * | 0.51 ** | 0.80 *** |
| B28 | Low veg cover (bare soil, rock outcrop) | 465 | −0.28 * | 0.67 ** | 0.95 ** |
| B35 | Lichen barren | 440 | −0.22 | 0.49 * | 0.71 ** |
| B36 | Lichen-shrub-herb-bare | 450 | −0.24 | 0.48 * | 0.72 ** |
| B38 | Rock outcrop, low vegetation cover | 450 | −0.51 ** | 0.81 *** | 1.32 *** |
| B41 | Low vegetation cover | 375 | −0.38 ** | 0.75 *** | 1.13 *** |

These phonological results corresponded well with the fall (i.e., September and October) and spring (i.e., May and June) temperature trends during the same period at the Inuvik climate station, located about 130 km east of the Ivvavik National Park. As Figure 4 shows, the fall temperature increased 2.63 °C during 1985–2013 and was significant at the 90% confidence level, while the spring temperature decreased slightly by 0.19 °C and was statistically not significant (Table 3).

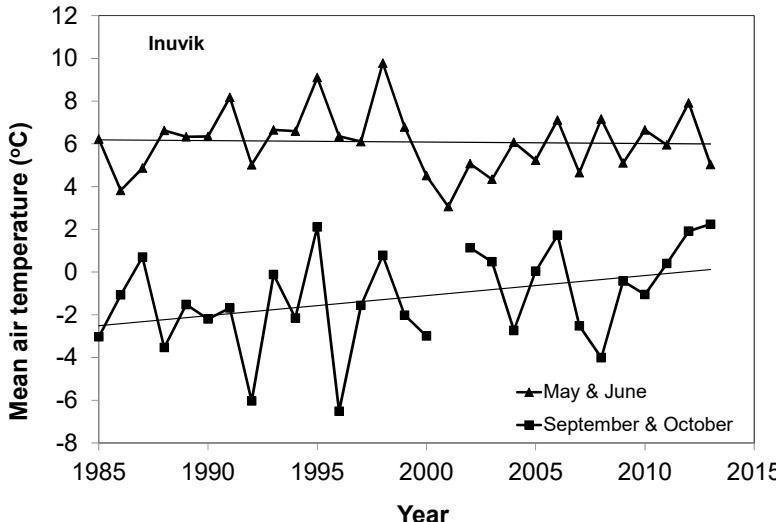

**Figure 4.** Inter-annual variations and trends in spring (May and June) and fall (September and October) mean temperatures at the Inuvik climate station in low-lying areas near the Ivvavik NP. Statistics for the two trends are listed in Table 3.

**Table 3.** Long term trends in spring (May and June) and fall (September and October) temperature at climate stations near the five study areas during 1985–2013. The trended change is the magnitude of change during 1985 and 2013, calculated as the value of the trend line in 2013—that in 1985.

| Climate Station | Study Area | Period | Trended Change (Days) | *p*-Value |
|---|---|---|---|---|
| Pond Inlet | Sirmilik | Spring | 1.12 | 0.297 |
| | | Fall | 3.33 | 0.013 |
| Churchill | Wapusk | Spring | 0.08 | 0.647 |
| | | Fall | 3.10 | 0.003 |
| Nain | Torngat | Spring | 1.45 | 0.056 |
| | | Fall | 2.11 | 0.001 |
| Inuvik | Ivvavik | Spring | −0.19 | 0.863 |
| | | Fall | 2.63 | 0.069 |
| Lupin | Bathurst | Spring | 1.50 | 0.145 |
| | | Fall | 2.73 | 0.011 |

The only statistically significant trend of the spring temperature during 1985–2013 was found at the Nain climate station, about 200 km south of the Torngat Mountains National Park, with an increase of 1.45 °C and significant at 90% confidence level. Correspondingly, SOS dates were advanced by a range of 5.5 to 14.6 days for various tundra classes in the Torngat Mountains National Park (Table 2). Although these SOS trends were statistically still not significant due to large inter-annual variations, they were substantial contributors to GSL increase compared with the trended changes in EOS for these tundra classes, ranging from 19 to 31.3 days. These results were consistent with the fact that the fall temperature at the Nain climate station was 2.11 °C and significant at the 99% confidence level.

### 3.2. Elevation Dependency of Trends in Plant Phenology

The long-term trends in SOS, EOS, and GSL could differ substantially within a relatively small study area (Table 2). These differences correlated well with the variations in elevation, as exemplified by Figure 5. For the 11 tundra classes in the Ivvavik National Park, 77% of the variation in the GSL trends during 1985–2013 could be explained by the elevation difference among classes, with $R^2 = 0.77$ and *p*-value = 0.0004. According to the relationship between GSL trend and elevation, we found a 55% increase in the GSL trend (i.e., from 0.58 d y$^{-1}$ to 0.89 d y$^{-1}$) over the 886 m elevation range from 26 m (the mean elevation of the lowest hedysar-avens inactive alluvial Terrace class) to 912 m (the mean elevation of the highest rock lichen class).

**Table 4.** Statistics for the relationships between phenology trend during 1985–2013 and elevation over the five study areas.

| Study Area | Phenology | Slope (d y$^{-1}$ m$^{-1}$) | Intercept (d y$^{-1}$) | $R^2$ | *p*-Value | *n* |
|---|---|---|---|---|---|---|
| | SOS | −0.000130 | −0.108 | 0.17 | 0.210 | 11 |
| Ivvavik | EOS | 0.000224 | 0.463 | 0.52 | 0.013 | 11 |
| | GSL | 0.000355 | 0.566 | 0.77 | 0.0004 | 11 |
| | SOS | −0.000660 | −0.119 | 0.65 | 0.016 | 8 |
| Sirmilik | EOS | 0.000617 | 0.210 | 0.59 | 0.025 | 8 |
| | GSL | 0.001308 | 0.322 | 0.71 | 0.008 | 8 |
| | SOS | −0.000300 | −0.272 | 0.48 | 0.057 | 8 |
| Torngat | EOS | 0.000387 | 0.764 | 0.40 | 0.092 | 8 |
| | GSL | 0.000746 | 1.025 | 0.63 | 0.019 | 8 |
| | SOS | −0.000170 | −0.296 | 0.001 | 0.944 | 6 |
| Wapusk | EOS | −0.001070 | 0.968 | 0.21 | 0.367 | 6 |
| | GSL | −0.000900 | 1.264 | 0.035 | 0.722 | 6 |
| | SOS | 0.000123 | −0.364 | 0.003 | 0.886 | 9 |
| Bathurst | EOS | 0.000911 | 0.206 | 0.087 | 0.441 | 9 |
| | GSL | 0.000789 | 0.570 | 0.025 | 0.683 | 9 |

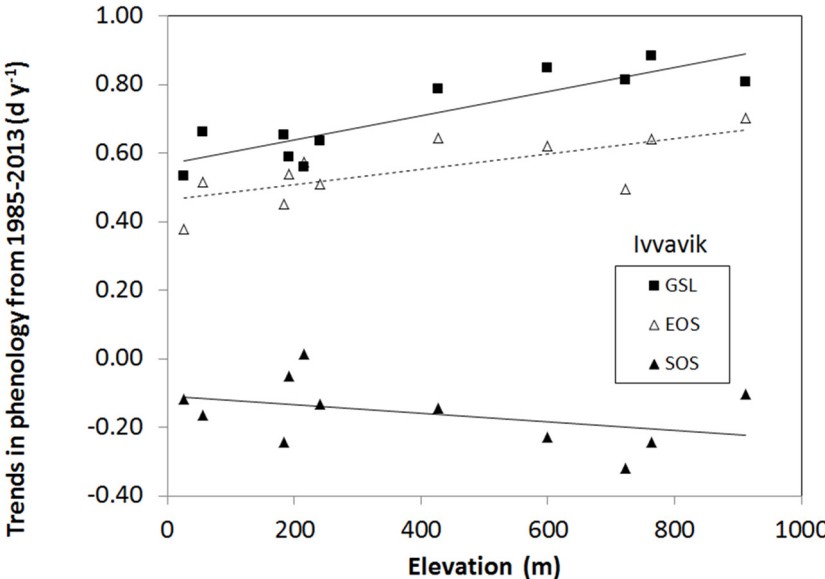

**Figure 5.** Relationships between SOS, EOS, or GSL trends (i.e., the slope of the trend line) during 1985–2013 and elevation for tundra classes in the Ivvavik National Park. Statistics for these relationships are listed in Table 4.

The elevation-induced lengthening of GSL in the Ivvavik National Park was mainly due to the further delay in EOS at higher elevations. Statistically, 52% of the variation in EOS trends during 1985–2013 among the 11 tundra classes in the Ivvavik National Park can be explained by their elevation differences, with $R^2 = 0.52$ and $p$-value = 0.012 (Figure 5). We estimated a 42% increase in EOS trend (i.e., from 0.47 d y$^{-1}$ to 0.67 d y$^{-1}$) from the relationship between EOS trends and elevation from elevation 26 m to 912 m. The change in EOS trends contributed 0.3 d y$^{-1}$ to the 0.41 d y$^{-1}$ increase in GSL trends over the elevation range from 26 m to 912 m. In comparison, the changes in SOS trends contributed only a 0.11 d y$^{-1}$ increase in GSL trends (Figure 5). Over the elevation range from 26 m to 912 m, we found that the SOS trend decreased from −0.11 d y$^{-1}$ to −0.22 d y$^{-1}$. However, we emphasized that the relationship between SOS trends and elevation was not statistically significant ($R^2 = 0.17$, $p$-value = 0.21).

The elevation dependency of EOS trends was further evidenced by the results over the Sirmilik and the Torngat Mountains National Parks (Figure 6). For the eight tundra classes in the Sirmilik National Park, elevation differences can explain 60% of the variation in EOS trends during 1985–2013, with $R^2 = 0.60$ and $p$-value = 0.025 (Table 4). From 208 m (the mean elevation of the lowest prostrate dwarf shrub class) to 611 m (the mean elevation of the highest barren class), the EOS trend increased 73%, namely, from 0.34 d y$^{-1}$ to 0.59 d y$^{-1}$. Similar results were also found for the Torngat Mountains National Park (Figure 6). Over the elevation range from the lowest shrub-herb-lichen-bare class at 23 m to the highest rock outcrop low vegetation cover class at 684 m, the EOS trend increased 33%, from 0.78 to 1.03 d y$^{-1}$, with $R^2 = 0.40$ and $p$-value = 0.092.

For the tundra classes in the Sirmilik and Torngat Mountains National Parks, significant elevation dependency was also found for the SOS trends (Figure 6 and Table 4). The elevation differences can explain 65% and 48% of the variations in the SOS trends during 1985–2013 among the 8 tundra classes in the Sirmilik National Park ($R^2 = 0.65$, $p$-value = 0.016), and Torngat Mountains National Park ($R^2 = 0.48$, $p$-value = 0.057), respectively.

With contributions from both effects of elevation on EOS and SOS trends, significant relationships between GSL trends and elevation were found for the tundra classes in the Sirmilik ($R^2 = 0.71$, $p$-value = 0.008) and Torngat Mountains National Parks ($R^2 = 0.63$, $p$-value = 0.019). Over the 403 m elevation range from the lowest class to the highest class in the Sirmilik National Park, the GSL trend increased 89%, from 0.59 to 1.12 d y$^{-1}$ during 1985–2013. Of the 0.53 d y$^{-1}$ elevation-induced increase in GSL trend, the effect of elevation

on EOS trend contributed 0.25 d y$^{-1}$, while that on SOS 0.27 d y$^{-1}$. Similarly, the 661 m difference in elevation from the lowest class to the highest class in the Torngat Mountains National Park increased the GSL trend by 47%, from 1.04 to 1.54 d y$^{-1}$ during 1985–2013. Of the 0.49 d y$^{-1}$ elevation-induced increase in GSL trends, 0.28 d y$^{-1}$ came from the effect of elevation on EOS trends, while 0.21 d y$^{-1}$ from the effect of elevation on SOS trends.

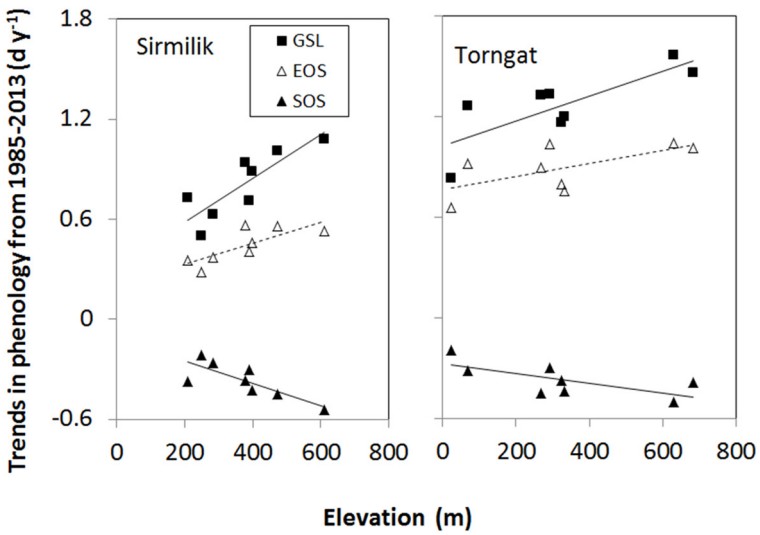

**Figure 6.** Relationships between SOS, EOS, or GSL trends during 1985–2013 and elevation for tundra classes in the Sirmilik NP and the Torngat Mountains NP. Statistics for these relationships are listed in Table 4.

## 4. Discussion

### 4.1. Hypotheses for Explaining the Elevation Dependency of Phenology Trends

Previous studies reported that in the Arctic and alpine regions, inter-annual changes and long-term trends of plant phenology were usually related to change in spring and fall temperatures [24,41–43]. At the same time, modification of snow accumulation on the ground could also play an important role. Therefore, we proposed two hypotheses regarding temperature change and snow accumulation to explain the elevation dependency of phenology trends.

The first hypothesis is that elevation in a mountain range might amplify the changes in spring and fall temperature (i.e., temperature amplifier hypothesis). As a result, spring or fall temperatures increased by higher rates at higher elevations, which in turn resulted in earlier SOS or delayed EOS at the higher elevations. Previous studies based on climate records have been inconclusive about this hypothesis, with supporting evidence e.g., [3–6] and negative results [17,18].

The second hypothesis is that an increase in plant growth, especially woody components, would enhance snow trapping and holding capacity (i.e., the biomass-snow hypothesis). A higher rate of woody biomass increase usually occurred at the lower elevations, which would likely trap more snow and increase snow depth on the ground in the spring [44–46]. The increased snow accumulation could then delay SOS compared to more exposed and lower albedo areas [43]. Borner et al. [21] examined the relationships between snow depths and phenology of four arctic plant species (*Betula nana*, *Salix pulchra*, *Eriophorum vaginatum*, and *Vaccinium vitis-idaea*) in arctic snow bed communities. They found that the SOS dates at the ambient sites were about seven days and two weeks earlier than at the mid and deep snow depth sites.

The temperature amplifier hypothesis can well explain the significant increases in EOS trends with elevation over all three mountainous study areas, as well as the SOS trends in the Sirmilik and Torngat National Parks (Figures 4 and 5, and Table 4). Significant increases in fall temperature at climate stations in low-lying locations near the three mountainous study areas have been observed, ranging from 2.11 °C near the Torngat Mountains National

Park to 3.33 °C near Sirmilik National Park (Table 3). According to this hypothesis, these fall temperature increases were further amplified at higher elevations, resulting in further delays in EOS. Similarly, spring temperatures increased by 1.12 °C at the Pond Inlet climate station near the Sirmilik National Park and 1.45°C at the Nain climate station near Torngat Mountains National Park (Table 3). The spring temperature increase was further amplified at a higher elevation, and so was the SOS advancing rate. The relationship between elevation and SOS in the Ivvavik National Park was insignificant (Table 4).

Nevertheless, this result can also be explained by the temperature amplifier hypothesis. There was essentially no change in spring temperature from 1985 to 2013 at the Inuvik climate station near the Ivvavik National Park, in fact, a slight decrease by 0.19 °C (Table 3). An amplified no-change at a higher elevation would still be a no-change. As a result, no significant increase in the SOS advancing rate with elevation was expected over the Ivvavik National Park.

The assessment of the biomass-snow hypothesis is more complicated. As Figure 7 shows, significant relationships between the trends of peak leaf biomass and SOS were found for classes in the Sirmilik National Park ($R^2$ = 0.84, $p$-value = 0.001, and $n$ = 8) and Torngat Mountains National Park ($R^2$ = 0.63, $p$-value = 0.018, and $n$ = 8) during 1985–2013. Here we used the trends in peak leaf biomass to represent the trends in woody biomass because the two are closely related to arctic tundra ecosystems [32,47]. At first glance, these results appear to suggest that the increase in peak leaf biomass could explain the decrease in the SOS advancing rate through the biomass-snow hypothesis for these two study areas. However, this hypothesis is likely invalid for the following reasons. (1) In the Ivvavik National Park, the opposite result was observed. The SOS advancing rates were higher for classes with a higher rate of increase in peak leaf biomass (Figure 7), instead of lower as predicted by the hypothesis. (2) Similarly, the increase in the peak leaf biomass rates also didn't significantly decrease the SOS advancing rates over the two flat study areas (Figure 7). (3) Physically, an increase in woody biomass could trap more snow but also accelerate the melting rate due to reduction in the albedo [45,48–50]. Albedo differences of 2–5% between high and low shrub cover landscapes had been detected from MODIS [49]. This difference corresponded to increased energy absorption of 3–6 W m$^{-2}$ [49], similar to the anticipated 3 W m$^{-2}$ forcing of summer anthropogenic warming [48]. (4) Finally, the trend of peak leaf biomass was also significantly correlated with elevation for tundra classes in the Sirmilik National Park ($R^2$ = 0.50, $p$-value = 0.05, and $n$ = 8) and Torngat Mountains National Park ($R^2$ = 0.75, $p$-value = 0.006, and $n$ = 8). The significant relationships between the trends of peak leaf biomass and SOS over the two mountainous study areas could be because both of their variations are the results of the elevation effect. In other words, it was the higher elevation and thus amplified spring temperature increase, instead of a reduction in woody biomass and less trapped snow, which caused the increase in SOS advancing rate.

While the biomass-snow hypothesis could not explain the aforementioned SOS trends over the two flat study areas, they can be explained well by the temperature amplifier hypothesis. Without many variations in elevations of different land cover classes over the two flat study areas, there was also no significant relationship between phenology trends and elevation (Figure 8, Table 4). Therefore, the phenology trends over the two flat study areas provided additional observational evidence supportive of the temperature amplifier hypothesis.

### 4.2. Causes That Could Mask the Elevation Dependency

In the upper panel of Figure 9, we plotted GSL trends against elevation for land cover classes in the five study areas. The pooled data from all five study areas showed no influence of elevation on the GSL trends, with $R^2$ = 0.0001, $p$-value = 0.94, and $n$ = 42. Even if the data were pooled only from the three mountainous study areas, as shown in the lower panel of Figure 9, the relationship between GSL trends and elevation was also weakened substantially to statistically not significant at the 90% confidence level, with $R^2$ = 0.103,

*p*-value = 0.102, *n* = 27. In comparison, the $R^2$ values ranged from 0.63 to 0.77 when a particular mountainous study area was concerned, with *p*-values being <0.05 in the case of Torngat Mountains National Park or <0.01 for the other two mountainous study areas. The primary reason for this masking effect is the difference in temperature change rates among different study areas. For example, the 2.63 °C increase in fall temperature at low-lying areas near the Ivvavik National Park from 1985 to 2013 was similar to the 2.11 °C increase near Torngat Mountains National Park. The spring temperature was slightly decreased by −0.19 °C near the Ivvavik National Park, in contrast to the 1.45 °C increase near the Torngat Mountains National Park. Consequently, the GSL trend of the highest class in the Ivvavik National Park (i.e., rock lichen class with a mean elevation of 912 m) was 0.81 d y$^{-1}$, slower than the lowest class in the Torngat Mountains National Park (i.e., the shrub-herb-lichen-bare class with a mean elevation of 23 m) at 0.84 d y$^{-1}$. These differences in temperature change rates among mountainous study areas could partially or even totally mask the elevation dependency.

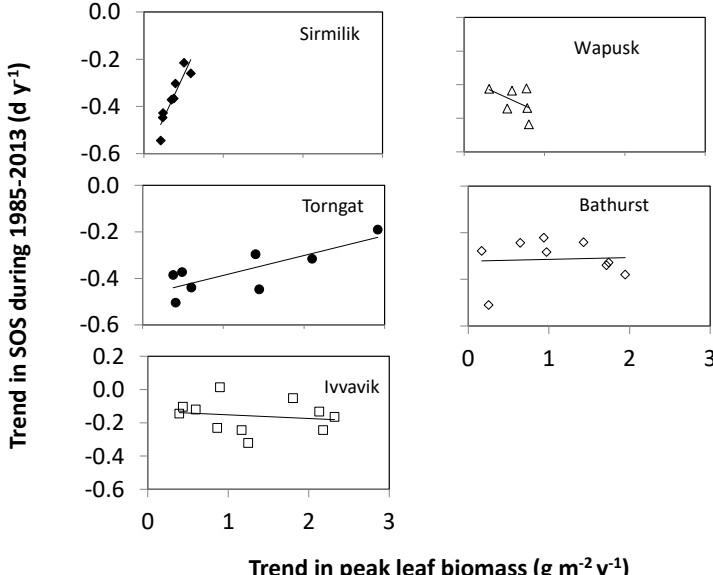

**Figure 7.** Relationships between trends in SOS and peak leaf biomass during 1985–2013 over all five study areas: Sirmilik ($R^2$ = 0.84, *p*-value = 0.001, *n* = 8), the Torngat Mountains ($R^2$ = 0.63, *p*-value = 0.018, *n* = 8), Ivvavik ($R^2$ = 0.03, *p*-value = 0.64, *n* = 11), and Wapusk National Parks ($R^2$ = 0.24, *p*-value = 0.33, *n* = 6), and the Bathurst caribou range ($R^2$ = 0.003, *p*-value = 0.89, *n* = 9).

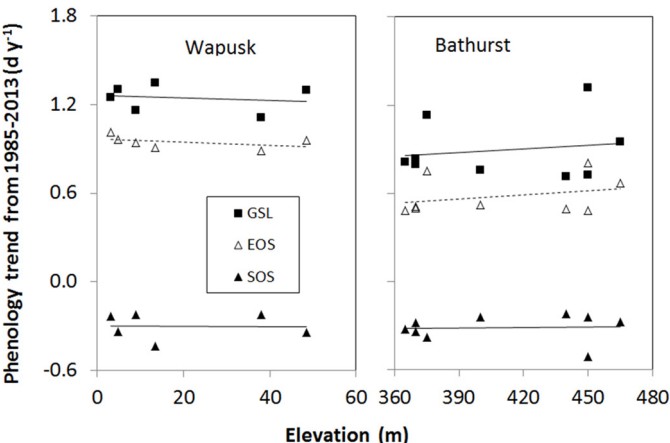

**Figure 8.** Relationships between SOS, EOS, or GSL trends during 1985–2013 and elevation over the two flat study areas: the Wapusk National Park and the Bathurst caribou summer range and calving ground. Statistics for these relationships are listed in Table 4.

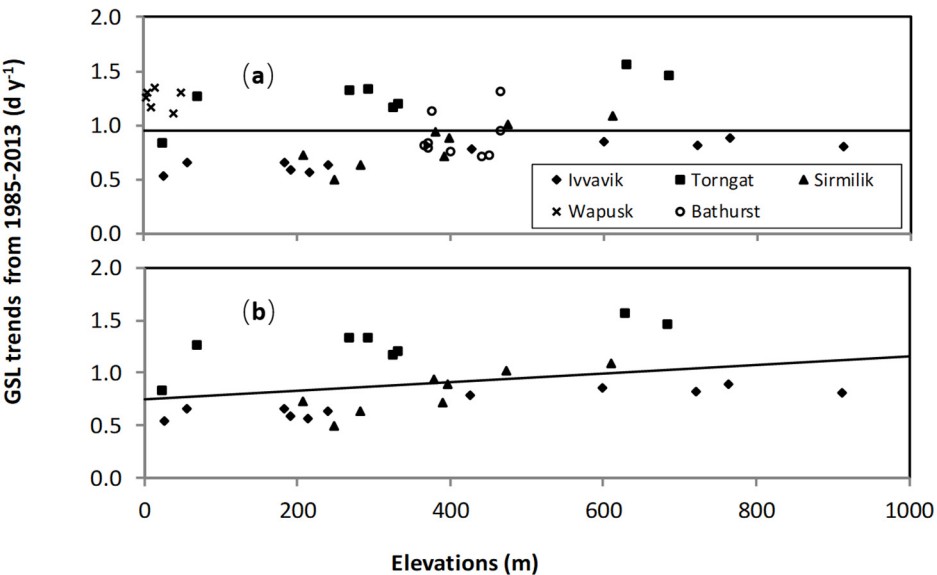

**Figure 9.** Relationships between GSL trends and elevation when data are pooled (**a**) upper panel: from all the five study areas in the upper panel ($R^2$ = 0.0001, *p*-value = 0.94, *n* = 42), and (**b**) lower panel: from all three mountainous study areas in the lower panel ($R^2$ = 0.103, *p*-value = 0.102, *n* = 27).

The approach of pooling data from study areas located in different climate zones regionally or globally is improper for investigating elevation dependency of temperature or phenology changes. Several previous regional or global scale studies (e.g., [13,14]) found no systematic relationship between the magnitude of temperature trends and elevation. We speculate that the data pooling might be one reason these studies didn't find the elevation dependency.

In addition, if there was no increasing trend in temperature at surrounding low-lying areas, then the amplification of elevation dependency would not result in an increased temperature trend at a higher elevation, as exemplified by the case of SOS trends over the Ivvavik National Park and the spring temperature change at the Inuvik climate station. Therefore, data from no temperature increase areas could also mask the elevation dependency.

## 5. Conclusions

We quantified SOS, EOS, and GSL for each of the tundra or wetland classes in the three mountainous study areas (i.e., Ivvavik, Sirmilik, and the Torngat Mountains National Parks), as well as the two relatively flat study areas (i.e., Wapusk National Park and Bathurst caribou range) from 1985 to 2013, using AVHRR time series. We came to the following conclusions based on trend analyses of these phenology dates and their relationships with other factors (e.g., elevation, peak leaf biomass, and spring and fall temperatures).

(1) The elevation dependency likely played an amplifier role upon a climate warming signal. When there was climate warming observed at weather stations in low-lying locations in a study area, we found significant increases in the magnitude of plant phenology change. For example, the magnitudes of long-term trends in EOS, SOS, and GSL increased significantly with elevation during 1985–2013 in all three mountainous study areas, except the SOS in the Ivvavik National Park.

(2) There was a similar variation of vegetation types from low-lying locations to high elevation for all three mountainous study areas. If the variation of vegetation types was the main driver of changes in the long-term trends of SOS, then we should also have found a similar increase of its magnitude with elevation in the Ivvavik National Park. However, there was no significant increase in the magnitude of the long-term trend in SOS from 1985 to 2013 with elevation in the Ivvavik National

Park. Correspondingly, the spring temperature at the Inuvik climate station located in the low-lying area of Ivvavik National Park decreased slightly by 0.2 °C from 1985 to 2013. Thus, long-term trends of SOS in the Ivvavik National Park indicated that the variation of vegetation types was not the primary driver of changes in the long-term trends of plant phenology. In addition, the peak leaf biomass values varied substantially for different land cover classes in the two flat study areas, similar to those in the three mountainous study areas. If the variation of vegetation types were the main driver of changes in plant phenology, then we would have detected a significant relationship in the long-term trends between the peak leaf biomass and plant phenology for all five study areas. However, we only found significant relationships in the long-term trends between the peak leaf biomass and SOS over the Sirmilik and the Torngat Mountains National Parks, but not the two flat study areas. Again, these results suggested that the variation of vegetation types was not the primary driver of changes in the long-term trends of plant phenology.

(3) As exemplified by the long-term trends of SOS and spring temperature in the Ivvavik National Park during 1985 and 2013, if there was no increase in temperature, then an amplified no-increase with elevation would still be no-increase. Therefore, when data are pooled from different climate zones regionally or globally, the elevation dependency can be partially or totally masked because of this no-warming or even cooling sites.

Because of their long-term and global coverages, the medium to coarse resolution satellite remote sensed data (e.g., AVHRR and MODIS) could provide an alternative data source for analyzing the elevational dependency of climate change. These remote sensing data could be especially useful for mountainous regions where long-term climate records are often unavailable. This study and other similar research suggest that high elevations could amplify climate warming, resulting in further accelerated glaciers melting and disappearing at high elevations. In considering the implications of an accelerated glacier disappearing from human society and the climate system, we urge more investigations into the elevation dependency of climate change.

**Author Contributions:** Conceptualization, W.C.; methodology, W.C.; software, L.W.; formal analysis, W.C. and L.W.; data curation, S.G.L., I.O. and R.L.; writing—original draft preparation, W.C; writing—review and editing, L.W., S.G.L., I.O. and R.L.; project administration, W.C.; funding acquisition, W.C. All authors have read and agreed to the published version of the manuscript.

**Funding:** The NWT Cumulative Impact Monitoring Program (CIMP), the Canadian Space Agency's Government Related Initiatives Program (GRIP), and the NRCan's Remote Sensing Science Program provided financial support for the study.

**Institutional Review Board Statement:** Not applicable. This study did not involve humans or animals.

**Informed Consent Statement:** Not applicable.

**Data Availability Statement:** The data presented in this study are available on request from the corresponding author.

**Acknowledgments:** Many northern students and residents (e.g., Alexander Gordon, Jayneta Pascal, and Kayla Arey from Aklavik, NWT; Simeonie and Jonas Johncena of Pond Inlet, Nunavut; Roy Judas and Brain Kodzin from Wekweeti) participated in the field measurements. Three anonymous reviewers provided constructive comments that greatly enhanced the manuscript. The authors want to thank all for their assistance.

**Conflicts of Interest:** The authors declare no conflict of interest.

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
