# Peer review of "Elevation-Dependent Changes to Plant Phenology in Canada’s Arctic Detected Using Long-Term Satellite Observations"

_atmosphere, doi:10.3390/atmos12091133_

Round 1

Reviewer 1 Report

The manuscript presents a good study of how changes in different phenological matrix is amplified by temperaure along elevation. I believev this kind of study is necessary and deserves to be consided for publication in Atmosphere. However, I do see several issues that need to be tackled before that, as desribed in the attached document. 

Author Response

Thank you for your very constructive comments. 

Reviewer 2 Report

This paper is very well prepared and its findings will be of great interest to a wide range of researchers. I suggest that the authors clarify the assumption of temperature measurements at one location/elevation is the same at all elevations associated with a given study area, i.e. no change in temperature with elevation despite the fact that higher elevations could potentially have cooler temperature for a given study site? This aspect of the study does not come through clearly in the manuscript. 

Author Response

Thanks for your constructive comments. We greatly appreciate your positive feedback.

Reviewer 3 Report

Chen et al. have examined whether vegetation exhibits long-term, elevation-dependent phenological responses to climate for 5 study areas in the Canadian Arctic. I commend the authors for addressing what is a very important climate-related issue.

Unfortunately, this paper needs major revisions particularly to the Introduction, Methods and Results section. I am particularly concerned that the methods do not follow best practice for extracting vegetation phenology metrics from satellite data, as well as the disparity in scale between the satellite data and field data. I believe if the authors can address these concerns, this paper would make a valuable contribution to climate change studies.

There are also numerous grammatical and spelling errors throughout the paper. The authors should consider using an independent proof-reader to address these.

Title

Consider changing the title to “Elevation-dependent changes to vegetation phenology in Canada’s Arctic detected using long-term satellite observations”

Abstract

The abstract is not well-structured e.g. the results (EOS, SOS etc) come before the methods (e.g. the years studied, study areas). The methods are vague e.g. what sensors/algorithms/field work were used? The authors use the term “elevation dependency” independently which is vague. Consider using “elevation-dependent response of vegetation”.  There is no mention of the significant discussion on the hypotheses for the observed trends in the results.

Introduction

Overall, the introduction is very poor primarily because the authors have not stated the proposed objectives or research questions, or the contribution this makes to understanding the very important issue of elevation-dependent responses to climate. The authors do not clearly describe how remote sensing of vegetation will be used as a proxy to determine if there are elevation-dependent changes in temperature. The authors should start with a more general paragraph introducing the broad climate-related issues facing the Arctic (as done in the abstract), and review any previous relevant work either for the Arctic, or where RS of vegetation phenology was used in this way. 

Materials and Methods

Overall, the methods are too short and confusing. This needs to be much more carefully written if a reviewer is able to determine whether the approach being used is robust. 

Study Area: Please provide a short description of the ecozones (i.e. what are the dominant vegetation communities) and whether these zones have been particularly impacted by climate change.

Data Sources: Define what SOS, EOS and GSL are. How are land classes defined? This is described further on but help the reader to understand this. 

Methods: The authors need to justify using the unbounded SRVI rather than the more widely used (and bounded) NDVI for vegetation phenology.

144: “The noise: ….” it is not clear what the authors are saying here.

147 - 150: It is not clear how the authors are ‘correcting’ for cloud contamination using a relationship with SRVI. The usual approach would be to exclude cloud affected pixels and gap-fill through an interpolation or modelling technique.

154 - 194: The author’s description of how they calculated SOS/EOS/thresholds from satellite data/ground data is extremely confused and unclear. This needs to be rewritten more clearly. The mixing of field data at a fine scale with coarse scale satellite data to derive phenological metrics might be problematic. The authors could have derived phenological metrics independently using the satellite data, and used the field data to validate the metrics derived. The date the field data was collected varied also across the years studied, and it is unclear how is was taken into account when using the field data with the satellite data.

It is also not good enough just to provide references to other papers rather than to describe explicitly what you are doing (e.g. the relationship described between biomass and SRVI starting line 189), and nor just to give a vague list of variables being tested (“e.g. mean elevation…” at line 194). 

I would suggest the authors break the methods down into more organised subsections e.g. retrievals of measurements from the field data, retrieval of phenology from the satellite data (and hoe the field data is used in this part), and finally the elevation-dependent modelling used to answer the objectives of the paper.

Results

Overall, the results are really unclear e.g. it is not clear whether the trends were derived from pixels for each landcover type within each study area. This is probably because the methods did not make this clear to me.

The first section of the results looks at long-term trends, and trends in temperature. The temperature trends could be in a separate sub-section.

The results in the first subsection seem to focus on one study area/class by providing the trend plots, but the other study areas are just included as tables? It is unclear why this particular area/class was the focus.

Table 2 is unclear. What does the number after the study are refer to? E.g. S10? And this column needs a name in the table.

The trends data (Tables 2,3) should give the actual statistical test values (e.g. R2/p-values). It is not clear what “Trends d y-1” in Table 2 is referring to. Is this the slope of the trend line? Same for “Trended change” in Table 3 - what does this mean?

Figure 5 y-axis - is this the slope of the trend line? If so, then use this terminology so it is more obvious what you are comparing with elevation. You use the term ‘slope’ in table 4 - is this the same as trend? And why are the units now d/y/m?

Discussion 

The discussion section is well-written compared to the previous sections. However, a lot of the background information here on the various proposed hypotheses could have appeared in the introduction   (see my comment above). I found that the methods and results section were poorly presented so that I don’t feel confident in commenting on whether the interpretations given here (and in the conclusion section) are robist and valid. Figures 7 - 9 should have been introduced in the results not in the discussion. 

Conclusion

1 - This first statement needs to be tempered - did elevation always play an amplifier role, or only in some cases? The rest of conclusion 1 then states there was an exception. The authors should be careful to only make conclusions for the areas/ecozones/years studied.

2 - It appears the authors are stating that there is always an elevation-dependent response of vegetation to elevation but that sometimes it is masked. The authors should temper this more by saying that possible dependencies may have been masked.  

3 - The authors didn’t explicitly state that investigating the effect of data pooling was an objective of this paper. The authors should include this in the introduction as an objective.

4 - The authors state that an increase in woody biomass “could” trap more snow/accelerate melting as a possible cause that impacts SOS, and are “likely” to cancel each other out, and the results confirmed this. The authors should be careful to distinguish their confirmed conclusions from observed results rather than what they hypothesise might be the underlying driver.

Final paragraph -these bigger picture questions should have been included in the introduction to provide content for the contribution given here. The authors should then end the conclusion on how the their study has contributed to answering these bigger questions.

Author Response

We appreciate your detailed and constructive comments. All your comments were addressed in a one-to-one format. 

Round 2

Reviewer 1 Report

The manuscript presents a good study of how different phenological matrix is amplified by temperature along elevation. I believe this kind of study is necessary, and deserves to be considered for publication in Atmosphere. The revised paper has greatly improved in terms of presenting the background and methods. I only have few minor comments.

Author Response

 The manuscript presents a good study of how different phenological matrix is amplified by temperature along elevation. I believe this kind of study is necessary and deserves to be considered for publication in Atmosphere. The revised paper has greatly improved in terms of presenting the background and methods.

Reply: We appreciated your constructive comments. They have played an important role in improving our MS.

I would recommend to

  • shorten the study area descriptions a little and leave the content that related to the study topic;

Reply: Thanks for the suggestion. We have shortened the study area descriptions.  

2) maybe simplify and clarify some language through the paper, for example, in 2.3.1, “we selected >50% purity as inclusion criteria”, I would think you might have the situation that A, B, C, D, and E composed of 40, 30, 10, 5, and 15% areas, then do you assigned the pixels as A?  Another example is that in conclusion, maybe merge point 2 and 3 since they are talking the same key point.

Reply: In the situation that A, B, C, D, and E are composed of 40, 30, 10, 5, and 15% areas, we didn’t use the pixel for analyzing plant phenology. Instead, we only selected the relative “pure” pixels for the analysis. Therefore, we added the following sentence to clarify the method. “Conversely, if all Landsat-derived classes within a 1-km AVHRR pixel were <50%, we would not include this pixel in the analyses.”   

As for merging the conclusion points 2 and 3, we revised as suggested.

Reviewer 3 Report

I commend the authors on the significant improvements made to their paper on this very important topic. I would like the the authors emphasise the importance and contribution of their research - see my comments below. I only have some other minor suggestions to improve the final paper.

Abstract

Line 29: “no warming”… this sentence is not clear

The abstract should include a concluding sentence that captures the important contribution of this paper. 

Introduction: 

Line 49: “For example “ - remove the word ‘famous’ from this sentence. This authors might consider removing this sentence as it defocuses the paper by including examples form other regions.

Line 53: “These wetlands….” - provide a citation for this sentence.

Line 56: “If the elevation…” - the Impact of climate change on glacial retreat is well studied - provide a regionally-relevant citation here.

Methods

The authors should review the methods to ensure they are clear and unambiguous so that a reader could independently reproduce this work.

Lines 275, 287, 289: change residue to residual (and check for other occurrences)

Lines 282 - 285: the authors will need to provide citations for linear/non-linear relationships of SRVI/NDVI.

Line 298: remove ‘developed and..’

Line 299: reword this final sentence to improve clarity.

Line 311: “overlapping areas’ correlations” - the authors will need to clarify what this means.

Line 319: “obtain their relationship” - the authors need to be more specific about this relationship and how it is obtained.

Line 312 - 328: this paragraph is long and difficult to follow. The authors should consider breaking this into two paragraphs.

Line 373: Reorder this sentence: ‘The relationship between the trends…. was investigated using regression”. Presumably the authors mean ‘linear regression’ here. The authors should be careful to state this throughout the paper so it is clear which type of regression is being used.

Results

Line 434: This new sentence needs to be reworded - it is still not clear how ‘trended change’ was calculated’

Conclusion

Line 719-722: Join these two sentences with If/Then e.g. “If the variation of vegetation types….., then we should also have found…”

The authors need to add a short concluding paragraph on the significance and impact of their study.

Author Response

Reviewer #3.

I commend the authors on the significant improvements made to their paper on this very important topic. I would like the the authors emphasise the importance and contribution of their research - see my comments below. I only have some other minor suggestions to improve the final paper.

Reply: We appreciated your constructive comments. They have played an important role in improving our MS.

Abstract

Line 29: “no warming”… this sentence is not clear

Reply: As suggested, we revised the words “no warming” to “no warming trend”.

The abstract should include a concluding sentence that captures the important contribution of this paper. 

Reply: We added the sentence to emphasize the importance of the study.  “This elevation-dependent climate change could have important implications for the fate of glaciers and ecosystems at high elevations under climate change.”

Introduction: 

Line 49: “For example “ - remove the word ‘famous’ from this sentence. This authors might consider removing this sentence as it defocuses the paper by including examples from other regions.

Reply: As suggested, we removed the sentence.

Line 53: “These wetlands….” - provide a citation for this sentence.

Reply: As suggested, we add a website that gives a detailed description of these wetlands: http://www.cen.ulaval.ca/bylot/en/bylotstudysite.php

Line 56: “If the elevation…” - the Impact of climate change on glacial retreat is well studied - provide a regionally-relevant citation here.

Reply: As suggested, we referenced the sentence to [7] and added a website that describes the impacts of climate change on glaciers. https://nsidc.org/cryosphere/glaciers/questions/climate.html

Methods

The authors should review the methods to ensure they are clear and unambiguous so that a reader could independently reproduce this work.

Reply: These methods have been reported in our previous papers ([24], [38-39]) in specialized remote sensing journals, such as Remote Sensing of Environment and the International Journal of Remote Sensing. In the paper, we can only give a brief decription of these methods. Readers who wish to reproduce this work over other regions are encouraged to reference our previous papers.     

Lines 275, 287, 289: change residue to residual (and check for other occurrences)

Reply: We revised them as suggested.

Lines 282 - 285: the authors will need to provide citations for linear/non-linear relationships of SRVI/NDVI.

Reply: We added citation [39] as suggested.

Line 298: remove ‘developed and..’

Reply: We revised them as suggested.

Line 299: reword this final sentence to improve clarity.

Reply: We revised the sentence as suggested.

Line 311: “overlapping areas’ correlations” - the authors will need to clarify what this means.

Reply: We added the following sentences to clarify. “There are usually overlapped areas between two adjacent Landsat Scenes. A correlation between pixels within the overlapped area could then be developed and corrected to the value of other scenes to that of the reference scene.”

Line 319: “obtain their relationship” - the authors need to be more specific about this relationship and how it is obtained.

Reply: We revised as suggested to the following sentence: “Linear regressions for red and near-infrared reflectances between Landsat and MODIS were used for the correction of Landsat-derived SRVI from the image acquisition date to the field measurement date.”

Line 312 - 328: this paragraph is long and difficult to follow. The authors should consider breaking this into two paragraphs.

Reply: We break the paragraph into two as suggested

Line 373: Reorder this sentence: ‘The relationship between the trends…. was investigated using regression”. Presumably the authors mean ‘linear regression’ here. The authors should be careful to state this throughout the paper so it is clear which type of regression is being used.

Reply: Yes, linear regression. We revised as suggested.

Results

Line 434: This new sentence needs to be reworded - it is still not clear how ‘trended change’ was calculated’

Reply: We revised the sentence as follows: “The trended change is the magnitude of change during 1985 and 2013, calculated as the value of the trend line in 2013 – that in 1985.”

Conclusion

Line 719-722: Join these two sentences with If/Then e.g. “If the variation of vegetation types….., then we should also have found…”

Reply: Revised as sugested.

The authors need to add a short concluding paragraph on the significance and impact of their study.

Reply: We added the following paragraph as suggested.

Because of their long-term and global coverages, the medium to coarse resolution satellite remote sensed data (e.g., AVHRR and MODIS) could provide an alternative data source for analyzing the elevational dependency of climate change. These remote sensing data could be especially useful for mountainous regions where long-term climate records are often unavailable. This study and other similar research suggest that high elevations could amplify climate warming, resulting in further accelerated glaciers melting and disappearing at high elevations. In considering the implications of an accelerated glacier disappearing to human society and the climate system, we urge more investigations into the elevation dependency of climate change.